# Zero-field *J*-spectroscopy of quadrupolar nuclei

**Román Picazo-Frutos** [1,2,3], **Kirill F. Sheberstov**[1,2,3,4], **John W. Blanchard** [1,2,3,5], **Erik Van Dyke**[1,2,3], **Moritz Reh**[6,7], **Tobias Sjoelander**[8,9,10], **Alexander Pines** [9,10], **Dmitry Budker** [1,2,3,6] **& Danila A. Barskiy** [1,2,3,9] ✉

Zero- to ultralow-field nuclear magnetic resonance (ZULF NMR) allows molecular structure elucidation via measurement of electron-mediated spin-spin *J*-couplings. This study examines zero-field *J*-spectra from molecules with quadrupolar nuclei, exemplified by solutions of various isotopologues of ammonium cations. The spectra reveal differences between various isotopologues upon extracting precise *J*-coupling values from pulse-acquire measurements. A primary isotope effect, $\triangle J = (\gamma_{^{14}N}/\gamma_{^{15}N})J_{^{15}NH} - J_{^{14}NH} \approx -58$ mHz, is deduced by analysis of the proton-nitrogen *J*-coupling ratios. This study points toward further experiments with symmetric cations containing quadrupolar nuclei, promising applications in biomedicine, energy storage, and benchmarking quantum chemistry calculations.

Zero- to ultralow-field nuclear magnetic resonance (ZULF NMR) is a variant of NMR in which measurements are performed in the absence of a large external magnetic field[1,2]. In such a regime (as opposed to conventional high-field NMR), intrinsic spin-spin interactions−*J*-couplings and dipole−dipole couplings−are not truncated by the coupling to the external magnetic field[3–5]. This condition opens a way for obtaining unique chemical information with modest instrumentation costs[6,7]. Because of its ability to detect subtle intrinsic spin-spin interactions that provide valuable information about chemical composition of the sample under study, ZULF NMR can serve as an advantageous detection modality in situations where the use of expensive superconducting magnets is undesirable or impossible[8]. In particular, the analysis of biologically-relevant samples (e.g., natural extracts or metabolites in bioreactors) using portable ZULF NMR sensors is a desirable goal[9,10]. In addition, applications of ZULF NMR include the search for axion-like dark matter and tabletop studies of physics beyond the standard model[11].

A typical ZULF NMR experiment consists of the following steps: (i) prepolarizing a sample in an external magnetic field, (ii) shuttling of the sample to the zero-magnetic-field region, (iii) applying magnetic field pulse(s) to the sample (or fast, non-adiabatic switching off of the guiding magnetic field) to generate a coherent nuclear spin evolution at zero field, and (iv) acquiring the NMR signal using a sensitive (5–10 fT/Hz^1/2) atomic magnetometer[5,6]. Processing of the signal is performed in a manner similar to high-field NMR experiments (i.e., via Fourier transformation)[12].

The first ZULF NMR measurements were performed on a handful of simple molecules containing a small number (2−5) of *J*-coupled nuclear spins. It was noted that in some cases ZULF NMR measurements did not result in observable *J*-spectra for substances under study[13]. Systems that do not produce observable ZULF NMR spectra typically contain spin-1 nuclei, such as deuterium (D), $^{14}$N, or $^{35}$Cl[14–16]. These nuclei are quadrupolar since, besides having a magnetic dipole moment, they possess electric quadrupole moment and, therefore, interact with the electric field gradients. For this reason, quadrupolar coupling typically dominates other nuclear spin interactions and causes, in the presence of molecular tumbling, fast relaxation on the order of few milliseconds (compared to seconds

[1]Helmholtz-Institut Mainz, 55099 Mainz, Germany. [2]Institute of Physics, Johannes Gutenberg-Universität Mainz, 55128 Mainz, Germany. [3]GSI Helmholtzzentrum für Schwerionenforschung GmbH, 64291 Darmstadt, Germany. [4]Department of Chemistry, École Normale Supérieure, PSL University, Paris, France. [5]Quantum Technology Center, University of Maryland, College Park, MD, USA. [6]Department of Physics, University of California—Berkeley, Berkeley, CA 94720, USA. [7]Kirchhoff-Institut für Physik, Universität Heidelberg, Im Neuenheimer Feld 227, 69120 Heidelberg, Germany. [8]Department of Physics, University of Basel, Klingelbergstrasse 82, Basel CH-4056, Switzerland. [9]Department of Chemistry, University of California, Berkeley, CA 94720-3220, USA. [10]Materials Science Division, Lawrence Berkeley National Laboratory, Berkeley, CA 94720-3220, USA. ✉e-mail: dbarskiy@uni-mainz.de

for spin-1/2 nuclei) leading to broadening of the ZULF NMR lines[12,17,18].

In this work, we systematically study the effect of quadrupolar nuclei on the observable ZULF NMR $J$-spectra. As a system for such study, we have chosen aqueous solutions of ammonium cations prepared in mixtures of $H_2O$ and $D_2O$ at high acidity (concentration of $H_2SO_4$ is ~2 M). We first demonstrate a $J$-spectra of [$^{14}$N]-ammonium ($^{14}NH_4^+$) and [$^{15}$N]-ammonium ($^{15}NH_4^+$) by taking advantage of the unique tetrahedral environment of the hydrogen atoms which effectively switches off quadrupolar interactions of the $^{14}$N nucleus (Fig. 1). Expensive isotopic labeling is not required for the preparation of these spin systems; this extends application opportunities for symmetric molecular ions in zero-field $J$-spectroscopy. We then report the most precise to date measurement of the $|J_{^{15}NH}/J_{^{14}NH}|$ ratio which we found to be in the range of $1.4009 - 1.4013$ (depending on specific peaks used for analysis) which is statistically different from $|\gamma_{^{15}N}/\gamma_{^{14}N}| = 1.4027$ listed in the table data for gyromagnetic ratios. Precision measurements of $J$-couplings performed on molecules in solution originating from the same container and unaffected by magnetic field fluctuations may be used as a valuable tool for benchmarking quantum chemistry calculations.

We then study the effect of deuterium by detecting $J$-spectra of $^{15}ND_xH_{4-x}^+$ (where $x = 0$, 1, 2, or 3). Mixtures of $^{15}ND_xH_{4-x}^+$ were prepared by dissolving $^{15}NH_4Cl$ in solutions with varying $D_2O/H_2O$ ratios and relying on rapid equilibration of isotopologues' concentrations due to chemical exchange between $H^+/D^+$ atoms[9].

As part of the study of $^{15}ND_xH_{4-x}^+$ ions using ZULF NMR $J$-spectra, we show that due to the hierarchy of nuclear spin-spin interactions (i.e., $|J_{^{15}NH}| > |J_{^{15}ND}| > |J_{DH}|$), different spectral lines in the $J$-spectra have different pulse-length dependences as demonstrated in experiment, rationalized by using perturbation theory, and confirmed by numerical simulations. The value of $-2.6(1)$ Hz for the D-$^1$H scalar coupling in $^{15}NDH_3^+$ is extracted from the experiment in which various spectral

lines in $J$-spectrum respond differently to magnetic field excitation pulses. We additionally discuss the possibility of creating and storing nuclear spin order in states with extended lifetimes which in principle should be achievable in the symmetric molecular cations presented in this work.

Based on results of this work, other symmetric molecular cations containing quadrupolar nuclei can be expected to produce well-resolved ZULF NMR $J$-spectra, with applications ranging from biomedicine to probing materials relevant to energy storage.

## Results and discussion

### Zero-field spectra of $^{14}$N- and $^{15}$N-ammonium isotopologues

To understand general features of $J$-spectra at zero field, we start from the analysis of the ZULF NMR spectrum of [$^{15}$N]-ammonium, a molecular ion that does not contain quadrupolar nuclei. A $J$-spectrum of [$^{15}$N]-ammonium ion consists of two lines, one at 110.114(9) Hz and another at 183.554(7) Hz provided by the heteronuclear spin-spin coupling, $J_{^{15}NH}$, between $^{15}$N and $^1$H spins (Fig. 1a). Particularly, the line at 110.114(9) Hz corresponds to $(3/2)|J_{^{15}NH}|$, where $J_{^{15}NH} = -73.42$ Hz is the $J$-coupling between $^{15}$N and $^1$H nuclei and it arises from the transitions between nuclear-spin energy levels with the total proton spin $K_A = 1$ (Fig. 1b). The line at 183.554(7) Hz corresponds to $(5/2)|J_{^{15}NH}|$ and arises from the transitions between states with a total proton spin $K_A = 2$. A corresponding high-field $^{15}$N NMR spectrum of the same sample would consist of five lines with intensities in the ratio of 1:4:6:4:1 separated by $J_{^{15}NH}$ as expected from the binomial distribution for the projections of the total proton spin on the magnetic field axis.

In contrast, ZULF NMR spectrum of [$^{14}$N]-ammonium ion exhibits three well-resolved lines at 52.4021(15) Hz, 104.7741(10) Hz, and 157.171(8) Hz (Fig. 1a). To explain this observation, one needs to consider five transitions, one in the $K_A = 0$ manifold, two in the $K_A = 1$ manifold, and two in the $K_A = 2$ manifold (Fig. 1c). Two pairs of transitions overlap resulting in three distinct peaks at $J_{^{14}NH}$, $2J_{^{14}NH}$ and $3J_{^{14}NH}$

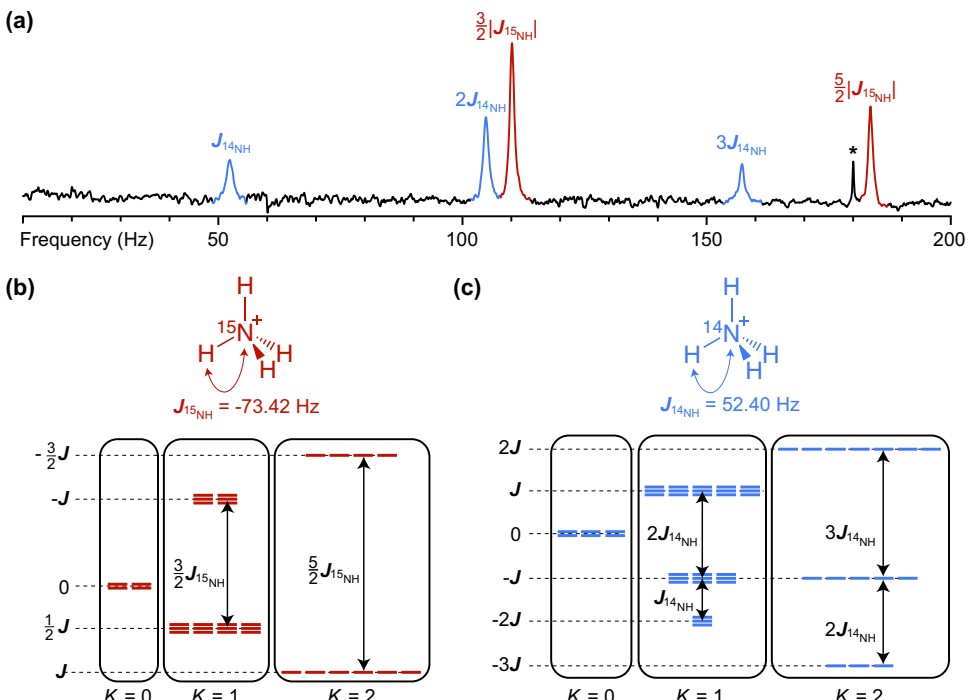

**Fig. 1 | $J$-spectra and energy levels of the ammonium isotopologues studied in this work. a** Zero-field NMR $J$-spectra of the mixture of [$^{15}$N]- and [$^{14}$N]-ammonium cations ($^{15}NH_4^+$ and $^{14}NH_4^+$) in aqueous solution of $H_2SO_4$ (2 M). The asterisk depicts the third harmonic of the 60 Hz power-line frequency. **b, c** Molecular diagram and nuclear spin energy-level structure of the [$^{15}$N]-ammonium and [$^{14}$N]-ammonium

cation, respectively. Observable transitions correspond to spin flips in the manifolds conserving the total proton spin $K_A = 0$, 1, and 2. The main interaction in the system is heteronuclear $J_{NH}$-coupling (in the simulation, we used $J_{HH} = 0$ since it does not affect transition frequencies and, thus, does not manifest in spectra).

in the ZULF NMR spectrum of $^{14}NH_4^+$ ($J_{^{14}NH} = 52.40$ Hz). Since $^{15}N$ and $^{14}N$ have different signs of gyromagnetic ratios, $^{15}NH_4^+$ and $^{14}NH_4^+$ have different signs of heteronuclear $J_{NH}$-couplings as indicated by inverted structures of their nuclear spin energy levels (Fig. 1b,c).

The mere fact of observing zero-field NMR $J$-spectrum of [$^{14}N$]-ammonium may seem surprising since $^{14}N$ is quadrupolar. Indeed, a typical nuclear spin Hamiltonian of a molecule containing quadrupolar nuclei is expected to have a large contribution arising from the interaction of the nuclear quadrupole moment with the electric field gradient created by the electrons at the nucleus. For this reason, quadrupolar nuclei typically exhibit fast relaxation on timescales on the order of a few milliseconds leading to their "self-decoupling". However, in the case of $^{14}NH_4^+$ cations, the $^{14}N$ is not self-decoupled because of the high symmetry of the molecule. In general, when quadrupolar nuclei are positioned at the center of a tetrahedral or an octahedral molecule, the quadrupolar interaction is effectively switched off since electric field gradients are absent in such symmetric environments. For the same reason, the quadrupolar tensor is typically averaged out in isotropic liquids by fast molecular motion[18]. Therefore, $^{14}NH_4^+$ and, as it will be seen below, $^{15}ND_xH_{4-x}^+$, are unique spin systems containing nuclei coupled to spins with $I=1$ which have relaxation times on the order of seconds. However, it is important to note that in general, quadrupolar nuclei can still affect relaxation of the coupled nuclear spins and therefore can lead to broadening of spectral lines in NMR spectra[18,19]. It is of little importance at high-field NMR as measurements of ammonium cations performed at 1 tesla yielded very similar proton $T_1$ values: $T_1(^{14}NH_4^+) = 3.22(2)$ s compared to $T_1(^{15}NH_4^+) = 3.25(2)$ s. In contrast, $^{15}N$ $T_1$ measured in the same field yielded 49(3) s, see Supporting Information (SI) for details.

**Precision measurement of $J$-couplings in $^{14}N$- and $^{15}N$-ammonium**
Indirect spin-spin coupling values between nuclei are mainly determined by the Fermi-contact interaction (i.e., probability of finding an electron inside a magnetic nucleus). Therefore, it is expected that the ratio of the experimentally measured $^{15}N - H$ and $^{14}N - H$ $J$-coupling values should be close to the ratio of the corresponding gyromagnetic ratios. A careful analysis revealed that the ratio of $J_{^{15}NH}$ and $J_{^{14}NH}$ coupling values differs from the ratio of the $^{15}N/^{14}N$ gyromagnetic ratios, and the difference is statistically significant. Specifically, from 36,000 measurements of the NH₄Cl sample containing 50:50 mixture of $^{15}N$ and $^{14}N$ isotopes, the ratio of the $J$-couplings ($|J_{^{15}NH}/J_{^{14}NH}|$) was determined to be within the range of $1.4009 - 1.4013$ (see Table 1) as compared to table data for gyromagnetic ratios $|\gamma_{^{15}N}/\gamma_{^{14}N}| = 1.4027548(5)$[20]. In the following discussion, we denote the three $J$-peaks of the $^{14}NH_4^+$ cation as ①, ②, and ③, and two peaks of the $^{15}NH_4^+$ cation as ▯ and ▯, for convenience.

One can see that depending on which peaks in the spectrum are used to estimate the $|J_{^{15}NH}/J_{^{14}NH}|$ ratio, the obtained values are different. However, a careful study of the systematic errors in the analysis procedure described below revealed that the values in Table 1 are consistent within the error bar. We note that although the peaks of $^{15}NH_4^+$ and $^{14}NH_4^+$ in ZULF NMR spectra are relatively broad (linewidths are ~1 Hz, determined by the proton exchange and not $T_2^*$), the precision of the center-frequency measurements is not limited by the linewidth[21]. To confirm if these error estimates indeed represent statistically significant variations in the measured values of $|J_{^{15}NH}/J_{^{14}NH}|$, we have run an additional analysis by constructing cumulative distribution functions (CDFs) using variable partitioning of the collected dataset (Fig. 2a).

Different partitions (i.e., accumulated groups of spectra) were constructed from the original data set, ranging from a single spectrum (the average of all 36,000 scans, Fig. 2b) to 3600 groups of averaged 10 scans. For each partition, the processing method described in the SI was carried out. By fitting spectral peaks using

**Table 1 | Ratios of $J_{NH}$-couplings for $^{15}NH_4^+$ and $^{14}NH_4^+$ extracted from the analysis of 36000 averaged spectra**

| ZULF NMR peaks used for the estimation | Measured $|J_{^{15}NH}/J_{^{14}NH}|$ value |
|---|---|
| (2/3) • (▯/①) | 1.4009(7) |
| (4/3) • (▯/②) | 1.40108(18) |
| 2 • (▯/③) | 1.40103(14) |
| (2/5) • (▯/①) | 1.4012(6) |
| (4/5) • (▯/②) | 1.40134(15) |
| (6/5) • (▯/③) | 1.40129(9) |

Lorentzian functions with four independent parameters (amplitude, peak center, phase, and linewidth), we extrapolated ratios of the $J$-coupling values as well as extracted the standard error of the fit; the processed dataset including the fits can be found in Table S2. Mean values and standard errors obtained directly from the analysis of the partitioned spectra are compared with the ones extracted from fitting the CDFs with equations for the sigmoid curve; the direct fit of the average of all 36,000 scans are indicated by the black dashed lines (Fig. 2c). We observed that measured values of the peak-center frequencies shift depending on the specific data processing protocol (Table S4, Fig. S3); the error bars shown in Fig. 2c represent the corresponding systematic errors. We find that the ratio of $J$-couplings determined from the spectra is significantly different from the ratio of gyromagnetic ratios (represented by the blue-dashed line)[20] irrespective of which peaks were used for analysis. Considering average values for the extracted $J_{NH}$-couplings, these differences are an order of magnitude larger than experimental uncertainty in the measured $|\gamma_{^{15}N}/\gamma_{^{14}N}|$ value, i.e., $\triangle J = (\gamma_{^{14}N}/\gamma_{^{15}N})J_{^{15}NH} - J_{^{14}NH} \approx -58$ mHz[20]. This is a manifestation of an isotope effect: there should be changes in the electronic structure of cations in solution that either affect Ramsey terms other than the Fermi-contact interaction[22], or the Fermi-contact interaction itself is affected by the changes in the vibronic structure of the cations caused by $^{15}N{\rightarrow}^{14}N$ isotope substitution[23-27]. This realization highlights unexplored opportunities for the use of ZULF NMR in precision measurements of spin-spin interactions. While similar analysis could in principle be performed using conventional high-field NMR, magnetic field drifts over long experimental time window and bounds on clock precision would necessitate additional post-processing which could introduce additional systematic errors while in ZULF NMR positions of the $J$-peaks are unaffected by field fluctuations and uncertainties associated with the demodulation of the reference clock.

While some analyses report precision measurements of $J$-coupling values using conventional NMR by considering a priori constant distance between the peak maxima[28], in our analysis no prior knowledge on the peak positions was assumed. Therefore, to the best of our knowledge, our study constitutes the first precision measurement of the $|J_{^{15}NH}/J_{^{14}NH}|$ ratio based on NMR spectroscopy unaffected by magnetic field fluctuations and additional signal post-processing to eliminate systematic errors.

Precision studies of $J$-couplings using zero-field NMR spectroscopy are proposed as a promising way to probe molecular chirality[29] and investigate parity non-conserving interactions in molecules[30]. In addition, precision measurements of molecular $J$-couplings in solution would allow comparing the results of calculating the precise molecular bond lengths by the tools of quantum chemistry with experiments, thus, making ZULF NMR a tool for benchmarking the precision of quantum chemistry calculations[31-33]. Further analysis using the modern tools of quantum chemistry could explain the exact dependence of $J$-couplings in ammonia on bond length (rotational and vibrational structure estimation). The presented measurements are relatively simple, and the investigated molecules are studied in liquid state and in the same container, thus, the effect of the dielectric properties of

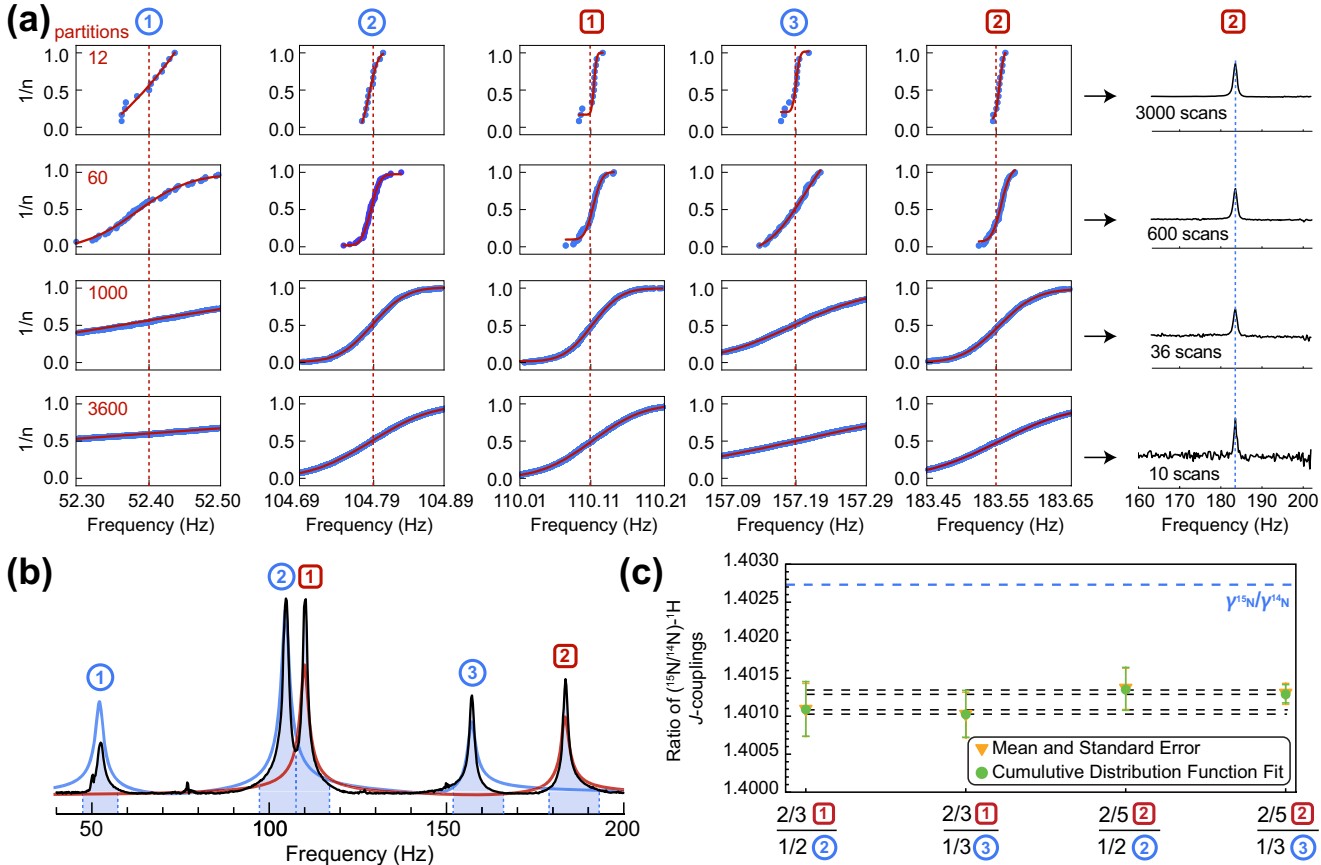

**Fig. 2 | Statistical analysis of the *J*-coupling ratio of the ammonium iso-topologues. a** Positions of the maxima extracted from peaks corresponding to isotopologues of ammonium organized in the ascending order (1/*n* where *n* is a partition number) generate cumulative distribution functions (CDFs); the averaged spectra with 3000, 600, 36, and 10 scans correspond to 12, 60, 1000, and 3600 partitions, respectively. **b** Zero-field NMR spectrum of [¹⁴N]- and [¹⁵N]-ammonium

(50:50 mixture, quadrature-detected magnitude mode, average of 36,000 scans) obtained with prepolarization at 2 T. **c** Values of the *J*-coupling ratios extracted from the analysis using 360 partitions (each represented by 100 averaged spectra). ZULF NMR peaks of the [¹⁴N]- and [¹⁵N]-isotopologues are labeled with blue circles and red squares, respectively.

the environment and temperature are accounted equally for both cations.

We note that a similar analysis could also be performed with high-field NMR. However, apart from the absence of the sensitivity to inhomogeneity and magnetic-field drifts, as well as the absence of stringent requirements on the reference clock, a major additional advantage of ZULF NMR is the absence of line shifts due to chemical exchange. Indeed, upon accelerated proton exchange ZULF NMR lines of ammonium broaden without changing their center frequencies while high-field NMR peaks move toward their "center of mass"[9,34].

**Zero-field spectra of Deuterium Isotopologues**

To analyze the effect on *J*-spectra of quadrupolar nuclei other than ¹⁴N, we performed ZULF NMR measurements of various isotopologues of ¹⁵ND$_x$H$_{4-x}^+$ (where $x$ = 0, 1, 2, or 3). While deuterium also has spin $I$ = 1 (so it is quadrupolar), its quadrupole moment is significantly smaller ($0.29 \cdot 10^{-30}$ m²) than that of ¹⁴N ($1.56 \cdot 10^{-30}$ m²)[18] and therefore its effect on ZULF NMR spectra is expected to be less pronounced. This expectation is now confirmed by the experiment as discussed below.

To prepare ¹⁵ND$_x$H$_{4-x}^+$ samples, the amount of deuterium was controlled by varying the ratio of H$_2$O/D$_2$O in the solution. Here and below, we refer to isotopologues as $x$D where $x$ represents a number of D atoms in the molecule (Fig. 3). Since chemical exchange leads to equilibration of various ammonium-isotopologue concentrations, a molar fraction $\chi$ of the isotopologue $x$D can be derived from the

fraction $p$ of the deuterium in solution as

$$\chi = \binom{4}{x} p^x (1-p)^{4-x}. \tag{1}$$

Figure 4a, b compares zero-field NMR spectra with high-field (18.8 T) ¹⁵N NMR spectra of samples containing ammonium chloride prepared by varying $p$, a fraction of deuterium in the solution. It is well known that ammonium ions undergo reversible hydrogen exchange with water[35]. High acidity of the samples can slow down the hydrogen exchange rate from ≈50,000 s⁻¹ down to ≈2 s⁻¹, thus, allowing for the molecule to exist long enough to exhibit nuclear spin coherences that can be detected with magnetometers in ZULF NMR spectroscopy[9].

Upon increase of the deuterium fraction ($p$), spectroscopic signatures of ¹⁵NH$_4^+$ gradually decrease; this is evident from both zero-field and high-field NMR spectra (Fig. 4a, b). However, these spectroscopic changes are more evident in ZULF NMR spectra because the absence of chemical shift simplifies the analysis of the spectra. While each of the chemicals has its own spectroscopic signatures characterized by a unique set of frequencies in *J*-spectra (Fig. 4c), high-field ¹⁵N NMR spectra of isotopologues suffer from the overlap of different multiplets and changes of the ¹⁵N chemical shift (often referred to as isotopic shift[25,36]), due to the changes in electronic density on the nitrogen atom upon increasing a number of deuterium atoms, $x$, from 0 to 4 (Fig. 4d).

**(a)**

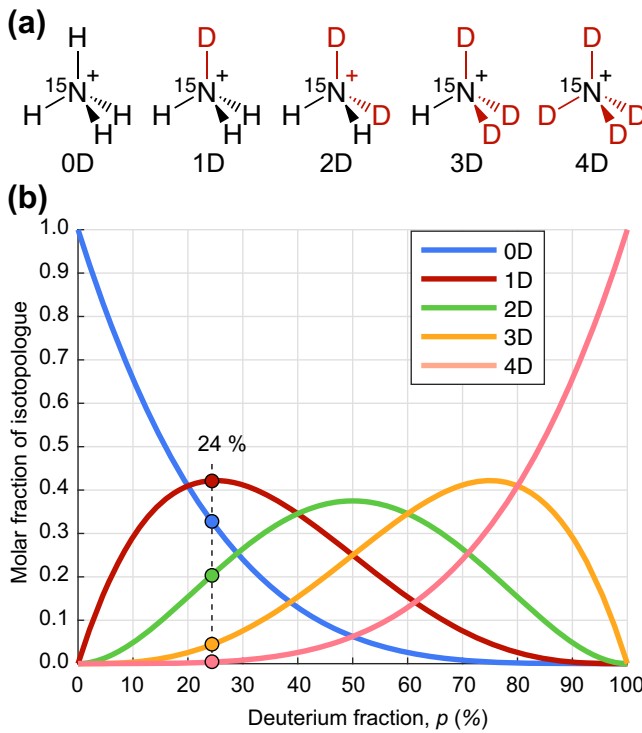

**(b)**

**Fig. 3 | The deuterium concentration in the samples studied in this work follows a binomial distribution. a** Isotopologues of the ammonium cation ($^{15}ND_xH_{4-x}^+$) are present in the aqueous solution. The fraction of deuterium was varied by changing the amount of $H_2O$ vs. $D_2O$. Isotopologues are labeled $x$D where $x$ represents a number of deuterium atoms in the molecule. **b** Equilibrium molar fractions of the isotopologues as a function of deuterium fraction in solution as determined by Eq. (1).

The main reason of the decreased ZULF NMR signals for ammonium cations with larger deuterium content is smaller initial magnetization of the sample due to decreased number of protons. Indeed, in a simple two-spin case, ZULF NMR signal is proportional to the difference between gyromagnetic ratios of the $J$-coupled nuclei[5]. In case of many-spin systems composed of more than two types of heteronuclei and prepolarization at high field, the detected signal depends on a combination of gyromagnetic ratios determined by the exact nuclear spin topology of the molecule[37]. However, the general trend is that the main magnetization-contributing nuclei, protons, predominantly contribute to the observed signal as they have significantly larger gyromagnetic ratio compared to other spins in the system, for example, $|\gamma_{^1H}/\gamma_{^{15}N}| \approx 10$ and $|\gamma_{^1H}/\gamma_{^2H}| \approx 6.5$.

The fact that the signal intensity in the $J$-spectrum of the molecule depends not only on the concentration but also on the exact spin-type composition and spin topology presents a general limitation of ZULF NMR spectroscopy for chemical analysis. This contrasts with the case of conventional high-field NMR where the detected signal is proportional to molecular concentration independent of topology[38]. Therefore, for analytical purposes, ZULF NMR spectral features of different chemicals should be calibrated (and/or simulated) for direct spectroscopic comparison. The agreement between the experimentally obtained and simulated ZULF NMR spectra is excellent for the $^{15}NH_4^+$, $^{15}NDH_3^+$, and $^{15}ND_2H_2^+$ molecules (Fig. 4). Signatures of other isotopologues $^{15}ND_xH_{4-x}^+$ (where $x$ is 3 and 4) are noticeable in the spectra as well, however, their signal is lower and, therefore, precise spectroscopic assignment is less straightforward.

We note that another possible reason of the decreased NMR signal for the molecules with larger deuterium content is scalar relaxation of the second kind (SR2K)[39]. In the context of ultralow-field NMR, SR2K

may lead to accelerated relaxation of overall polarization as well as coherences involving spin-½ nuclei mediated through $J$-couplings with the quadrupolar nucleus. Accelerated relaxation is especially pronounced in the regime where nuclei are strongly coupled[40]. In simple words, when spins are strongly coupled, they tend to relax together rather than separately, and the presence of a quadrupolar spin relaxing on the timescale of milliseconds acts as a relaxation sink for the rest of the spin system. For example, fast liquid-state polarization decay was observed for hyperpolarized [5-$^{13}$C]-glutamine during the transfer to an MRI scanner when the transfer field was below 800 µT and was attributed to the SR2K caused by the fast-relaxing quadrupolar $^{14}$N-nucleus adjacent to the $^{13}$C nucleus in the amide group[41]. Another study showed that partially deuterated ethanols have significantly larger linewidth in near-zero-field Larmor-precession experiments implying a strong SR2K contribution to $^1$H relaxation rates despite relatively weak coupling constants on the order of 1−2 Hz[14]. However, the situation is different for $^{15}ND_xH_{4-x}^+$ isotopologues investigated in our work because, first, we do observe their zero-field $J$-spectra and, second, the linewidth of resonances is determined by the intermolecular chemical exchange and not the SR2K mechanism. The absence of an overwhelming SR2K contribution may potentially be explained by the fact that the quadrupolar relaxation rates of the spin state imbalances in $^{15}ND_xH_{4-x}^+$ are determined by the correlation time of the molecular rotations and thus, symmetry properties of these molecules must be accounted for. An exact explanation of the relaxation phenomena at zero field requires additional analysis which lies beyond the scope of this paper. We note that long-lived spin states in molecules containing deuterium, were demonstrated for the deuterated methyl groups[42].

As a general rule, $J$-spectra featuring couplings to quadrupolar nuclei should be observable at ZULF conditions if the corresponding $J$-coupling is manifested in conventional high-field NMR spectra. For example, as seen from both Fig. 4a, b, a splitting of the NMR lines due to $J_{^{15}ND}$ is clearly observed in the zero-field and high-field spectra.

## Evaluation of $J_{HD}$ from ZULF NMR spectra

High-field $^1$H NMR measurements cannot directly resolve $^1$H-D $J$-coupling in $^{15}ND_xH_{4-x}^+$ systems. For example, Fig. 5a shows a $^1$H NMR spectrum of the [$^{15}$N]-ammonium solution with a 24% fraction of deuterium atoms. One can clearly see a series of doublets split by heteronuclear $^1$H-$^{15}$N $J$-coupling of −73.4 Hz shifted to lower field for each isotopologue $^{15}ND_xH_{4-x}^+$ as $x$ increases from 0 to 2; however, the fine structure corresponding to $^1$H-D coupling is not visible and only broadening of the peaks is observed. To investigate opportunities of resolving $^1$H-D $J$-coupling with zero-field NMR techniques, we numerically simulated zero-field spectra for deuterated ammonium isotopologues as well as analyzed their energy-level structures using the perturbation theory (see SI).

For all isotopologues $^{15}ND_xH_{4-x}^+$ (where $x = 1 - 3$) one can distinguish regions of low-frequency (0−30 Hz) peaks and high-frequency peaks (120−170 Hz). High-frequency peaks correspond to transitions within the strongly coupled subsystem consisting of $^{15}$N and $^1$H spins when total deuterium spin remains unchanged. Low-frequency peaks correspond to the deuterium spin flips where the strongly coupled $^{15}$N-$^1$H subsystem remains unperturbed. Due to the hierarchy of interactions in the molecule ($|J_{^{15}NH}| > |J_{^{15}ND}| > |J_{HD}|$) these groups of peaks are expected to respond differently to the magnetic pulse excitation and thus, by plotting their integrals as a function of the pulse length, one can extract information about subtle spin-spin interactions which otherwise would have taken more sophisticated multinuclear high-field NMR analyses (Fig. S4). By performing such analysis, we were able to constrain the value of $J_{HD}$ to −2.6(1) Hz.

Interestingly, unlike in the case of high-field NMR, signs of $J$-couplings alone can significantly affect the positions of the ZULF NMR

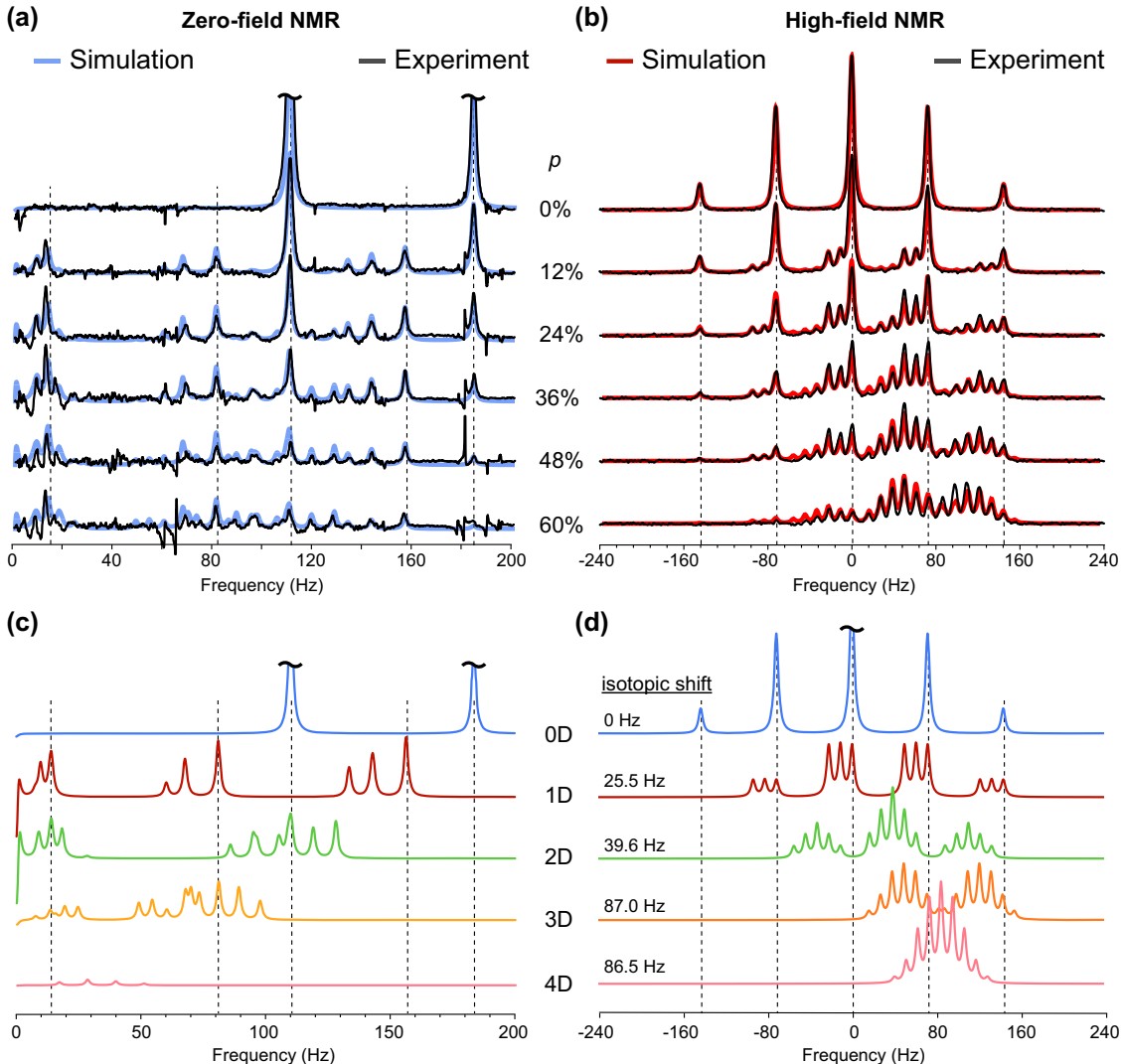

**Fig. 4 | Zero- and high-field spectra of ammonium isotopologues with varying deuterium concentration.** Experimentally measured (**a**) zero-field NMR spectra and (**b**) high-field (18.8 T) $^{15}$N NMR spectra of $H_2O/D_2O$ solutions containing a mixture of isotopologues of ammonium, $^{15}ND_xH_{4-x}^+$, prepared by varying the molar fraction of deuterium, $p$. Simulated NMR spectra for mixtures of the individual isotopologues calculated using weighting factors from Eq. (1) are overlaid. Simulated $J$-spectra of individual isotopologues for zero-field and high-field $^{15}$N NMR spectra are shown in (**c**) and (**d**), respectively.

peaks (Fig. 5b). Using a sample with deuterium fraction of 24% as an example, we simulated the $J$-spectra by varying signs of $J_{^{15}NH}$, $J_{^{15}ND}$, and $J_{HD}$ (one should note that reversal of signs for all spin-spin couplings at the same time would result in the same spectral pattern, however, the value of $J_{^{15}NH}$ is known to be negative, see Fig. 1). Especially pronounced are effects of the $J$-coupling sign change on the intensity of the high-frequency peaks of $^{15}NDH_3^+$ as well as on the nutation pattern for the low-frequency peaks.

Numerical calculations support the experiments which demonstrate that high-frequency and low-frequency peaks have maximal intensity at different pulse lengths. This ability to maximize some parts of the $J$-spectra while suppressing others opens possibilities for on-demand spectral editing in zero-field NMR spectroscopy using duration of the excitation pulse alone. Indeed, in case when complex chemical mixtures are studied using ZULF NMR tools, this feature could yield an additional degree of freedom for disentangling the spectral complexity. In principle, the experiment can be performed in a two-dimensional (2D) manner where Fourier transformation in the indirect (pulse length) dimension would give separate peaks in the 2D spectrum revealing additional information about spin topology that may simplify molecular characterization. We note that experiments with

$^{14}ND_xH_{4-x}^+$ could also be carried out, additionally avoiding isotopic enrichment with $^{15}$N nuclei. However, since their ZULF NMR spectra would have additional splittings due to the presence of spin-1 $^{14}$N nucleus, SNR would be further reduced making the analysis more time consuming.

Symmetric ions such as $^{14}NH_4^+$ and $^{15}NH_4^+$ represent an interesting class of molecules for storage of the nuclear spin order. Indeed, less symmetric analogs of ammonium, methyl groups, are known to support long-lived spin states (LLSS)[43,44] and long-lived coherences (LLC)[45] due to the fact that transitions between the states of different irreducible representations are symmetry-forbidden for intramolecular dipole-dipole relaxation[42]. While LLSS have already been observed in $^{13}CH_3/^{13}CD_3$ groups[42,46,47], the search of spin orders with extended lifetimes in $^{14}NH_4^+/^{15}NH_4^+$ is certainly warranted. The ability to store nuclear spin coherences for a long time will enable enhanced spectral resolution (i.e., narrow lines) in zero-field NMR. In addition, the existence of the LLSS in symmetric ions could allow preserving hyperpolarization on the timescale significantly longer than $T_1$. One should note that hyperpolarization of ammonia and its derivatives was demonstrated by using signal amplification by reversible exchange (SABRE) approach[48,49].

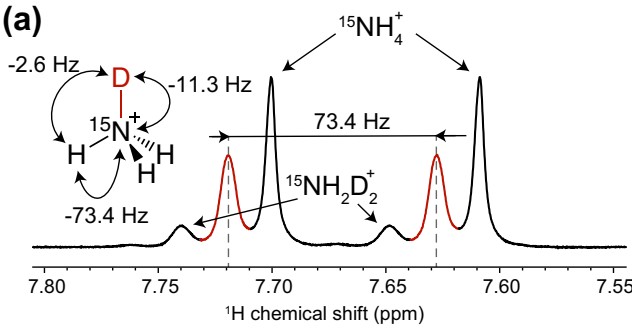

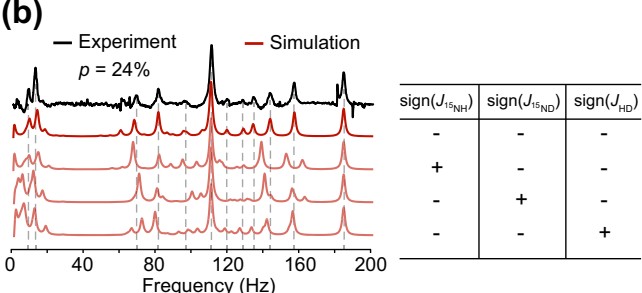

**Fig. 5 | Zero-field NMR spectroscopy allows the direct extraction of the sign of *J*-couplings. a** $^1$H NMR spectrum (800 MHz) and (**b**) ZULF NMR spectrum (black) of $^{15}ND_xH_{4-x}^+$ solution with deuterium fraction $p = 24\%$. In (**b**), calculated zero-field *J*-spectra of a mixture of isotopologues is shown in red, corresponding signs of *J*-couplings are presented on the right. Given insensitivity to global sign changes, only the four non-redundant permutations from the total eight possible combinations of *J*-coupling signs are shown.

Other molecules containing quadrupolar nuclei in tetrahedral or octahedral environments may serve as subjects for future investigation by the ZULF NMR spectroscopy. For example, $BF_4^-$ and $PF_6^-$ are counter-ions in various organometallic complexes and ionic liquids, as well as electrolytes in batteries or electrochemical double-layer capacitors[50]. Thus, zero-field *J*-spectroscopy of symmetric ions including ones with quadrupolar nuclei can find potential application for probing Bronsted acid sites in zeolites, lipid bilayers, and the hydrated ionomer membranes used in fuel cells[51].

Some molecules containing quadrupolar nuclei may find applications in biomedicine. For example, choline and its derivates are proposed as biomarkers for cancer diagnosis and response to treatment using hyperpolarized in vivo NMR spectroscopy[52,53]. While most of the research to date is focused on polarizing $^{15}$N-labeled choline derivatives, based on the findings of this study, preparation of $^{14}$N hyperpolarization in non-labeled (i.e., $^{14}$N-containing) molecules should be feasible as well. While $T_1$ of $^{14}$N sites in choline is still relatively short (~4 s)[54], which may limit applicability of the conventional approach, the long-lived spin states can provide a platform for the storage of enhanced population imbalance. At the same time, ZULF NMR detection of the characteristic spectroscopic signatures does not require hyperpolarization and therefore, small molecules containing $^{14}$N in semi-symmetric environments can be used as tracers for in vivo ZULF NMR studies. For example, acetylcholine is an important neurotransmitter and monitoring its concentration by optical magnetometers could yield valuable metabolic information[55]. Feasibility for the realization of these ideas is supported by the fact that $^1$H–$^{14}$N heteronuclear correlation (HSQC) spectroscopy of choline-containing compounds in solutions has been successfully demonstrated in vitro[54] and in vivo (i.e., "rule of thumb" for quadrupolar nuclei is fulfilled)[56,57].

To summarize, we report zero-field NMR measurements of molecules featuring the *J*-coupling to quadrupolar nuclei. Solutions containing different isotopologues of ammonium cations, $^{14}NH_4^+$ and

$^{15}ND_xH_{4-x}^+$ (where $x = 0$, 1, 2, or 3) were studied and their zero-field NMR *J*-spectra were measured. Molecules containing a larger number of deuterons compared to protons are characterized by a lower intensity of resonances in *J*-spectra as attributed to the decreased number of protons in the molecule (less overall magnetization) and not the scalar relaxation of the second kind. For the spin systems containing more than two types of magnetic nuclei, different groups of peaks in *J*-spectra have a different dependence of the magnetic pulse length provided a suitable hierarchy of nuclear spin-spin interactions ($|J_{^{15}NH}| > |J_{^{15}ND}| > |J_{HD}|$). Spin-spin coupling values and their signs are extracted for $^{15}NDH_3^+$ and $J_{HD}$ value was determined to be −2.6(1) Hz. First to date precision measurement of $|J_{^{15}NH}/J_{^{14}NH}|$ is performed and the value differs within a range of $1.4009 - 1.4013$. The statistically significant difference of this value with the measured ratio of $^{15}N/^{14}N$ gyromagnetic ratios is reported, this primary isotope effect is attributed to the changes in electronic structure and rovibronic energy potential for the cations. Such subtle differences in electronic structure for $^{15}NH_4^+$ and $^{14}NH_4^+$ are manifested via *J*-spectra and for the first time are reported for the molecules having identical environment in the liquid phase. Detection of ammonium ions by atomic magnetometers at zero field provides specificity toward various isotopologues and paves the way for studying other symmetric molecules containing quadrupolar nuclei with potential applications in biomedicine and energy storage.

## Methods
### Experimental apparatus
Zero-field *J*-spectra of the ammonium samples (Figs. 1, 4–5, and S4) were recorded using a home-built ZULF NMR spectrometer (incorporating $^{87}$Rb atomic magnetometer inside multilayer magnetic shielding). Physical principles of operation, construction, and calibration of the instrument are described in detail in ref. 6. Each *J*-spectrum is a result of 100 averages, polarization time is 30 s in the field of 2 T, shuttling time to the zero-field chamber is ~0.5 s. During the shuttling, no guiding field was applied. To generate an observable ZULF NMR signal, a magnetic pulse of variable length was applied in a direction of the magnetometer sensitive axis.

The data presented in Fig. 2 was recorded using a home-built ZULF NMR spectrometer with multilayer magnetic shielding to attenuate Earth's magnetic field $10^6$-fold based on commercial sensors[58,59] (QuSpin QZFM Gen-2; $4 \times 4 \times 4$ mm$^3$ Rb vapor cell) with a gradiometric configuration[60]. A total of 36,000 scans were acquired, with 6 s polarization time at 1.8 T. The shuttling time to the zero-field chamber is ~0.5 s. The measurement protocol consisted of the following steps: (i) a piercing solenoid was used to apply 100 μT field for 0.4 s during the sample shuttling (using a pneumatic setup); (ii) during the same period, an additional 100 μT in the shuttling direction was also applied using a Helmholtz-coil pair; (iii) during 0.1 s, the piercing solenoid was switched off while the Helmholtz field remained (the previous two steps account for the shuttling time of 0.5 s); (iv) to ensure adiabaticity towards the sensitive axis (orthogonal to the shuttling direction) we performed a field "crossing", where the field of 100 μT was ramped down to zero within 50 ms while simultaneously a different Helmholtz coil along the magnetometer sensitive axis ramped the field from zero to 100 μT; (iv) finally, the field was non-adiabatically (<10 μs) switched off, resulting in an observable signal.

### Sample preparations
Solutions of $^{15}NH_4Cl$ (Sigma-Aldrich 299251) and/or $^{14}NH_4Cl$ (Sigma-Aldrich 254134) were prepared in a 6 M concentration by dissolving them in de-ionized water followed by the addition of 98% (mass percent) sulfuric acid (Sigma-Aldrich) to a final concentration of 1.9 M. Solutions were then purged of dissolved oxygen by bubbling with nitrogen gas for 10 min at 1 atm. The samples were then flame sealed in

5 mm NMR tubes under vacuum. Sample volumes were 300 μL for ZULF measurements and 500 μL for high-field measurements.

Solutions of $^{15}ND_xH_{4-x}^+$ were prepared by dissolving $^{15}NH_4Cl$ (320 mg per vial) into a mixture of distilled water and $D_2O$ (Cambridge Isotope Laboratories DLM-4-99.8-1000) according to a desired deuterium fraction ($p$) followed by the addition of concentrated sulfuric acid (98% by mass) to yield a total volume of 1.0 mL with 6 M $^{15}NH_4Cl$ and 1.9 M $H_2SO_4$. 300 μL portions of solution were transferred to standard 5 mm NMR tubes and subsequently sealed with a flame.

### Numerical and analytical calculations of NMR spectra

Zero-field spectra of $^{14}NH_4^+$ and $^{15}ND_xH_{4-x}^+$ were calculated using a code written in Wolfram Mathematica and Python (see supplementary files "Supplementary Software 1", "Supplementary Software 2", "Supplementary Data 3"). The initial state of the density operator corresponded to spins being polarized along the $z$-axis according to Boltzmann distribution considering the gyromagnetic ratio of each nucleus. Liouville-Von-Neumann equation was solved numerically to calculate the evolution of the density matrix under the action of zero-field Hamiltonian. The observable operator corresponded to the total magnetization produced by all the spins along the $z$-axis. The time-dependent density matrix was projected onto the observable operator to provide the time domain signal, which was then Fourier transformed to produce a zero-field spectrum for a given isotopomer. Additionally, an analytical perturbation model was used (see SI) to calculate eigenfrequencies and eigenstates of the zero-field Hamiltonian for each of the isotopomers of ammonia cation and to cross-check numerical calculation and to gain a better understanding of underlying physics in the studied molecular systems.

### Data availability

Fitting data is completely defined by the parameters reported in Tables S1-S4. The file "Supplementary Data 3" contains the averaged of all scans in text format (.lvm).

### Code availability

Simulations of ZULF-NMR spectra and analysis are provided in the supplementary material as well as in separate supplementary files: "Supplementary Software 1", "Supplementary Software 2", "Supplementary Data 1", "Supplementary Data 2".

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

## Acknowledgements

We acknowledge the financial support by the Alexander von Humboldt Foundation in the framework of the Sofja Kovalevskaja Award. D.B. acknowledges support by the Cluster of Excellence Precision Physics, Fundamental Interactions, and Structure of Matter (PRISMA + EXC 2118/1) funded by the DFG within the German Excellence Strategy (Project ID 39083149).

## Author contributions

D.A.B. conceptualized research and wrote the first draft of the manuscript. R.P.F., K.F.S., E.V.D. updated the main manuscript and the Supplementary Material. R.P.F., J.W.B., M.R., E.V.D., K.F.S., D.A.B. collected data. R.P.F., K.F.S., J.W.B., T.S., A.P., D.B., D.A.B. contributed to the analysis and interpretation. D.B. and D.A.B. supervised the research. All authors edited and approved the final manuscript.

## Funding

## Competing interests

The authors declare no competing interests.
