## [Peer Review File · Nature Communications]

Zero-field J -spectroscopy of quadrupolar nucleiREVIEWER COMMENTS

Reviewer #1 (Remarks to the Author):

Can the authors comment on the following points?

- 1) At what B₀ field were the T₁ measurements performed? How different would the values be if measured at 'the other' field?
- 2) I visually estimate a FWHH for the spectrum of Fig. 1 of ca 2-3 Hz. How has the uncertainty of the reported J(15NH)/J(14NH) estimated? Shouldn't this be related to the experimental linewidth?
- 3) Considering that the simulated spectra shown in color in Fig. 3A are always pure absorptive, what is the origin of the dispersive-like peaks in these spectra? Why do these latter become more important with the increase of deuteration?
- 4) How well do the experimental spectra of Fig. 3 follow the relative intensities predicted by Fig. 2?
- 5) How well do the experimental nutations of Fig. 5B match those predicted by the theory?
- 6) The 'conclusions' section begins with a discussion about LLS which seems a bit out of place considering that LLS are never mentioned before in the manuscript. It almost seems like it should be moved in the 'introduction' part.
- 7) What are the limitations of the technique if applied to 14NDxH4-x+? If one advantage offered by this experimental approach is that of avoiding isotopic enrichment then this point should be clarified.

Reviewer #2 (Remarks to the Author):

In their paper "Zero-Field J-spectroscopy of Quadrupolar Nuclei" the authors, pioneers of ZULF NMR, present a concise study of Zero-Field NMR of small molecules bearing quadrupolar nuclei. The authors show that J spectra of the isotopologues 14NH₄⁺, and 15NH_{4-x}D_x⁺ can indeed be observed. Overall the paper is very well written, and the reproducibility should be excellent. As argued by the authors, the chosen, highly symmetric ammonium cations are expected to show rich relaxation behaviour which will likely lead to further studies.

The authors claim that the low intensity of deuterium-bearing ammonium cations is due to less overall magnetization, and not due to scalar relaxation of the second kind. The authors say that simulations of the spectra are detailed in the appendix. The appendix describes the energy structure and transitions of the ammonium cations in great detail, but there is no description of how the spectra are simulated. Are they merely superpositions of Lorentzians at the calculated transition frequencies? Or is there a quantitative understanding of the signal intensity for each isotopologue? If so, how do the experimental intensities compare to the theoretical predictions? Related to these issues I am not convinced by the comparison of theory and experimental data in Figure 3. For example, the authors overlay the theoretical spectrum for NH₂D₂ (maximum intensity at 50% deuteration) with the experimental data at 36% deuteration, and the theoretical spectrum for NHD₃ (maximum intensity near 75% deuteration) with the experimental data at 48% deuteration. If the authors understand the intensity of the spectra I would suggest strongly to compare the experimental data with correctly weighted averages of the theoretical spectra for x=1,2,3,4. This would enable a qualified comparison of experimental data and theoretical predictions, and also a qualified comparison of ZULF and high-field NMR data. The (simulated) spectra of the individual isotopologues (e.g., NH₃D) could then be shown in the appendix. I also note that the noise level seems to be higher in the 60% spectrum compared to the 0% spectrum. Has the same scaling been applied to all experimental spectra?

In any case the authors should state clearly whether the signal intensities are understood.

If (as I hope) yes, the authors should simply state that SR2K is not required to explain the signal intensities, and list the coefficients for the various isotopologues in the appendix.

If not, the linewidth does put an upper bound on relaxation due to SR2K during the acquisition, as the authors argue. However, the authors should then discuss potential relaxation during the

sample transfer where the Larmor frequency might match the deuterium T1. There is also the possibility of so-called slow SR2K (Elliott et al., doi:10.1063/1.5074199) The authors may consider publishing the experimental raw data and/or the experimentally determined weighting factors that best match the experimental data (the data probably allow a determination of these factors only for $x = 0, 1, 2$). If the intensities are not understood, then I believe it would still be possible to measure the relaxation of the different isotopologues via a variable delay prior to the magnetic pulse. Such a measurement would immediately quantify any potential SR2K at zero field and be a valuable addition to the manuscript.

My second main criticism refers to the value of the ratio for the $1\text{H}-15\text{N}$ and $1\text{H}-14\text{N}$ J couplings, reported as 1.4012(2). The discrepancy w.r.t the ratio of the respective gyromagnetic ratios is attributed to the secondary isotope effect. The reported level of precision deserves a bit of added scrutiny. The authors need to detail the error analysis, and how they arrive at the error margin. If a linear-least squares algorithm has been used, then I believe the error analysis would only hold for a residuum dominated by white noise. I believe that the reported value is based on the data shown in Fig. 1A (a reference should be added to the text). Upon close inspection, in Fig. 1A, the theoretical curves at $2J_{\{14\text{N}-\text{H}\}}$ and $3J_{\{14\text{N}-\text{H}\}}$ appear at a slightly higher frequency than the experimental curves. Given this discrepancy, what would convince me of a significant secondary isotope effect would be a plot of the residual for (i) the reported value of $J_{14\text{NH}}$, and for (ii) a calculated value of $J_{14\text{NH}}$ based simply on the ratio of gyromagnetic ratios, and ignoring the secondary isotope effect.

Further points:

Related to the scope of the work, the authors claim that ZULF NMR spectroscopy of ammonium cations can be applied for probing materials relevant in biology and energy storage and/or where expensive superconducting magnets are not available. However, it seems that ZULF requires high (typically molar) concentrations of analytes. Due to the low sensitivity, the competing technology seems to be table-top NMR with permanent magnets, which indeed has become a widespread analysis modality in recent years. Also, the statement that expensive labeling is not necessary is a bit questionable given the extensive use of 15N in the present study. My personal view is that ZULF NMR has a rather long way to go before it can be used in applications for biology and energy storage. That said, the present study is a nice example of the rich set of phenomena in physical chemistry that may be studied with ZULF NMR.

There is a statement on page 12 that each chemical has its own spectroscopic signatures characterized by a unique set of frequencies in the J spectrum, compared to high-field 15N NMR spectra that would suffer from overlap of different multiplets, and changes in 15N chemical shift. This statement almost suggests that the ZULF NMR is more capable here than the high-field NMR. I believe that the chemical shifts and J couplings could readily be obtained even from the 1D NMR data sets. The authors should also detail the acquisition parameters of the NMR data in the appendix. I expect that high-field NMR would furthermore reveal the 15ND_4 spectrum, further simplifying analysis.

With respect to the pulse-length analysis, details of how the dependence is calculated should be added to the appendix. In Fig. 5, how are the black data points in panel A calculated from the spectra shown in Fig. 5B? For example the 15NH_4 peak at 110 Hz 10 us seems to be bigger than the (inverted) peak at 15 us, yet the data points in panel A indicate 1.5 and -1.5, respectively.

In the conclusion there is a statement that 14N in semi-symmetric environments can be used as tracer for /in vivo/ ZULF NMR studies without hyperpolarization. Even if the complete blood volume of an animal was replaced with a concentrated solution of ammonium chloride the filling fraction / signal would be only one tenth of what the authors get now. So how could this work?

Minor points / suggestions::

p. 4: It is very clear from the appendix what the authors mean by "the hierarchy of spin-spin interactions", but it would probably help the reader to explain this concept in the main text.

"We extract $15\text{N}-1\text{H}$, $14\text{N}-1\text{H}$ spin-spin coupling values : replace comma with and

p. 5: add articles various spectral lines in /the/ J-spectrum, ... We additionally discuss /the/ possibility

p.6 We start from the analysis of the ZULF NMR spectrum of [¹⁵N]-ammonium /cation/

p. 9 /in/ the same container

p. 12 ... upon increasing /the/ number of deuterium atoms

p. 12 ... Indeed, in a simple two-spin case, /the/ ZULF NMR signal...

p. 24 I have not checked the tables in detail, but I think the transition $3/2, 1$ to $1/2, 1$ should be moved down by one line in table 2. In the header of table A6 the Eigenstate symbols have to be changed to F and K_B

I am very much looking forward to see the authors' response and future work, as well as to learning where I was wrong! I wish them all the best for the future.

Benno Meier

Reviewers' comments:

Reviewer #1 (Remarks to the Author):

Can the authors comment on the following points?

1) At what B0 field were the T1 measurements performed?

Authors' response: Since we found errors in the presentation of T_1 values, we have run additional benchtop NMR measurements. Now proton T_1 values for [^{14}N]- and [^{15}N]-ammonia as well as T_1 of ^{15}N in [^{15}N]-ammonium are presented in Figure S4 of the Supporting Information.

Changes made to the manuscript: We added T_1 values to the following sentence

"...measurements of ammonium cations performed at 1 tesla yielded very similar proton T_1 values: $T_1(^{14}\text{NH}_4^+) = 3.22(2)$ s compared to $T_1(^{15}\text{NH}_4^+) = 3.25(2)$ s. An ^{15}N T_1 measured in the same field yielded 49(3) s, see Supporting Information (SI) for details."

How different would the values be if measured at 'the other' field?

Authors' response: For small molecules such as NH_4^+ , rotational correlation time is typically on the timescale of hundreds of ps, thus, the extreme narrowing regime applies and relaxation time is determined by $J(0)$, i.e., spectral density function at zero frequency, for an achievable range of magnetic fields. Since the system is symmetric, quadrupolar and chemical shift anisotropy (CSA) contributions to relaxation can be neglected for all types of nuclei, and the main relaxation mechanism at play is dipole-dipole (DD) relaxation. For this reason, we expect individual relaxation times of nuclei in NH_4^+ to be approximately the same at various fields.

2) I visually estimate a FWHH for the spectrum of Fig. 1 of ca 2-3 Hz. How has the uncertainty of the reported $J(^{15}\text{NH})/J(^{14}\text{NH})$ estimated? Shouldn't this be related to the experimental linewidth?

Authors' response: We thank the referee for the comment. Experimental linewidth in ZULF NMR spectra of $^{14}\text{NH}_4^+$ and $^{15}\text{NH}_4^+$ is determined by chemical exchange. Nonetheless, precision of the center frequency determination is not determined by the linewidth. It is known that in cases when NMR peaks are well separated (difference between peaks' center frequencies is much larger than their linewidth), and the signal line-shape is Lorentzian, the precision is determined by the signal-to-noise ratio [Jupp et al. *J. Magn. Reson.*, 1998] which is the case in our study. To make sure the presented ratio of J -couplings is correct, we have run additional measurements (36000 scans) and a comprehensive error analysis, added corresponding figures and sections in the text. In addition, ZULF NMR measurements allow extracting center frequencies corresponding to different J_{NH} irrespective of chemical exchange (i.e., upon accelerated proton exchange ZULF NMR lines of ammonium broaden without changing their center frequencies while high-field NMR peaks move toward their "center of mass").

Changes made to the manuscript: We cite relevant literature and updated the text in multiple places throughout the manuscript.

3) Considering that the simulated spectra shown in color in Fig. 3A are always pure absorptive, what is the origin of the dispersive-like peaks in these spectra? Why do these latter become more important with the increase of deuteration?

Authors' response: This is a good question. We indeed observed somewhat dispersive-like character of the near-zero-frequency lines for partially deuterated isotopologues. The origin of the dispersive character lies more likely in the chemical exchange effects as well as details of the signal processing which may affect near-zero-frequency peaks. A detailed investigation of the origin of this effect lies beyond the scope of the paper, however, the text was modified to remove accent from the nutation curve corresponding to the low-frequency peaks.

5) How well do the experimental spectra of Fig. 3 follow the relative intensities predicted by Fig. 2?

Authors' response: After receiving constructive criticism from the Reviewers, we have run additional measurements and completely changes the mentioned figures. Now the agreement between simulations and the experiment is excellent.

Changes made to the manuscript: Multiple changes made to all figures in the manuscript.

6) How well do the experimental nutations of Fig. 5B match those predicted by the theory?

Authors' response: We moved the discussion concerning nutation effects to the Supporting Information, however, point out that experimental nutation curves match those predicted by theory reasonable well.

Changes made to the manuscript: Experiments discussing nutation effects were moved to the Supporting Information.

7) The 'conclusions' section begins with a discussion about LLS which seems a bit out of place considering that LLS are never mentioned before in the manuscript. It almost seems like it should be moved in the 'introduction' part.

Authors' response: We thank Referee for the comment. We modified the outlook section now is rewritten to reduce the emphasis on LLS.

Changes made to the manuscript: Outlook section was modified.

7) What are the limitations of the technique if applied to $^{14}\text{ND}_x\text{H}_{4-x}^+$? If one advantage offered by this experimental approach is that of avoiding isotopic enrichment then this point should be clarified.

Authors' reply: We agree with the comment on avoiding isotopic enrichment even further.

Changes to the manuscript: The following sentence is added to the text

"We note that experiments with $^{14}\text{ND}_x\text{H}_{4-x}^+$ could also be carried out, additionally avoiding isotopic enrichment with ^{15}N nuclei. However, since their ZULF NMR spectra would have additional splittings due to the presence of spin-1 ^{14}N nucleus, SNR would be further reduced making the analysis more time consuming."

Reviewer #2 (Remarks to the Author):

In their paper "Zero-Field J-spectroscopy of Quadrupolar Nuclei" the authors, pioneers of ZULF NMR, present a concise study of Zero-Field NMR of small molecules bearing quadrupolar nuclei. The authors show that J spectra of the isotopologues $^{14}\text{NH}_4^+$, and $^{15}\text{NH}_{4-x}\text{D}_x^+$ can indeed be observed. Overall the paper is very well written, and the reproducibility should be excellent. As argued by the authors, the chosen, highly symmetric ammonium cations are expected to show rich relaxation behaviour which will likely lead to

further studies.

Authors' reply: We thank the referee for the positive assessment of the quality of science in the manuscript.

The authors claim that the low intensity of deuterium-bearing ammonium cations is due to less overall magnetization, and not due to scalar relaxation of the second kind. The authors say that simulations of the spectra are detailed in the appendix. The Supporting Information describes the energy structure and transitions of the ammonium cations in great detail, but there is no description of how the spectra are simulated. Are they merely superpositions of Lorentzians at the calculated transition frequencies? Or is there a quantitative understanding of the signal intensity for each isotopologue? If so, how do the experimental intensities compare to the theoretical predictions?

Authors' reply: We now add an additional Mathematica file as a Supporting Information describing in detail simulation of the ZULF NMR spectra.

Changes to the manuscript: Multiple changes including in-depth Supporting Information.

Related to these issues I am not convinced by the comparison of theory and experimental data in Figure 3. For example, the authors overlay the theoretical spectrum for NH₂D₂ (maximum intensity at 50% deuteration) with the experimental data at 36% deuteration, and the theoretical spectrum for NHD₃ (maximum intensity near 75% deuteration) with the experimental data at 48% deuteration. If the authors understand the intensity of the spectra I would suggest strongly to compare the experimental data with correctly weighted averages of the theoretical spectra for x=1,2,3,4. This would enable a qualified comparison of experimental data and theoretical predictions, and also a qualified comparison of ZULF and high-field NMR data. The (simulated) spectra of the individual isotopologues (e.g., NH₃D) could then be shown in the appendix. I also note that the noise level seems to be higher in the 60% spectrum compared to the 0% spectrum. Has the same scaling been applied to all experimental spectra? In any case the authors should state clearly whether the signal intensities are understood.

Authors' reply: We thank the Referee for the suggestion. It is now performed, and experimental data is overlaid with simulations containing correctly weighted spectra of all isotopologues.

Changes to the manuscript: Multiple changes including new Figures in the main text and Supporting Information.

If (as I hope) yes, the authors should simply state that SR2K is not required to explain the signal intensities, and list the coefficients for the various isotopologues in the appendix. If not, the linewidth does put an upper bound on relaxation due to SR2K during the acquisition, as the authors argue. However, the authors should then discuss potential relaxation during the sample transfer where the Larmor frequency might match the deuterium T₁. There is also the possibility of so-called slow SR2K (Elliott et al., doi:10.1063/1.5074199) The authors may consider publishing the experimental raw data and/or the experimentally determined weighting factors that best match the experimental data (the data probably allow a determination of these factors only for x = 0, 1, 2). If the intensities are not understood, then I believe it would still be possible to measure the relaxation of the different isotopologues via a variable delay prior to the magnetic pulse. Such a measurement would immediately quantify any potential SR2K at zero field and be a valuable addition to the manuscript.

Authors' reply: We thank Referee for the suggestions. New Figure 4 clearly demonstrates that changes in intensity are due to magnetization and not SR2K. The discussion is modified accordingly.

My second main criticism refers to the value of the ratio for the ^1H - ^{15}N and ^1H - ^{14}N J couplings, reported as 1.4012(2). The discrepancy w.r.t the ratio of the respective gyromagnetic ratios is attributed to the secondary isotope effect. The reported level of precision deserves a bit of added scrutiny. The authors need to detail the error analysis, and how they arrive at the error margin.

Authors' reply: We thank the Referee. This comment prompted us to perform a series of additional measurements and data analysis. While confirming our original conclusions, the current manuscript is a major improvement and in-depth expansion of the original work.

Changes to the manuscript: Multiple changes and a completely new analysis is now presented in Figure 2.

If a linear-least squares algorithm has been used, then I believe the error analysis would only hold for a residuum dominated by white noise. I believe that the reported value is based on the data shown in Fig. 1A (a reference should be added to the text). Upon close inspection, in Fig. 1A, the theoretical curves at $2J_{\{^{14}\text{N-H}\}}$ and $3J_{\{^{14}\text{N-H}\}}$ appear at a slightly higher frequency than the experimental curves. Given this discrepancy, what would convince me of a significant secondary isotope effect would be a plot of the residual for (i) the reported value of $J_{^{14}\text{NH}}$, and for (ii) a calculated value of $J_{^{14}\text{NH}}$ based simply on the ratio of gyromagnetic ratios, and ignoring the secondary isotope effect.

Authors' reply: We agree, we have rerun the whole analysis using cumulative distribution functions (CDFs). The mere fact that CDFs follows sigmoid curves indicates that the noise in the vicinity of the studied peaks is white and normal statistical analysis is applicable. We also avoid terminology of "secondary" isotope effect which was not applicable to the situation in question.

Changes to the manuscript: Multiple changes throughout the text.

Further points:

Related to the scope of the work, the authors claim that ZULF NMR spectroscopy of ammonium cations can be applied for probing materials relevant in biology and energy storage and/or where expensive superconducting magnets are not available.

However, it seems that ZULF requires high (typically molar) concentrations of analytes. Due to the low sensitivity, the competing technology seems to be table-top NMR with permanent magnets, which indeed has become a widespread analysis modality in recent years.

Authors' reply: We overall agree with the Referee. However, we would like to point out that ZULF NMR does not suffer from magnetic field drifts affecting the measurement results which obviating the need of additional field-lock and/or sophisticated post-processing for taking them into account.

Also, the statement that expensive labeling is not necessary is a bit questionable given the extensive use of ^{15}N in the present study.

Authors' reply: We agree, the text was rewritten.

Changes to the manuscript: Now the sentence in question reads:

“Detection of ammonium ions by atomic magnetometers at zero field provides specificity toward various isotopologues and paves the way for studying other symmetric molecules containing quadrupolar nuclei with potential applications in biomedicine and energy storage.”

My personal view is that ZULF NMR has a rather long way to go before it can be used in applications for biology and energy storage. That said, the present study is a nice example of the rich set of phenomena in physical chemistry that may be studied with ZULF NMR.

Authors' reply: We agree, we rewrote the sentence and removed those speculations completely. This is why we are excited to be in a pioneering phase of this work.

There is a statement on page 12 that each chemical has its own spectroscopic signatures characterized by a unique set of frequencies in the J spectrum, compared to high-field ¹⁵N NMR spectra that would suffer from overlap of different multiplets, and changes in ¹⁵N chemical shift. This statement almost suggests that the ZULF NMR is more capable here than the high-field NMR. I believe that the chemical shifts and J couplings could readily be obtained even from the 1D NMR data sets. The authors should also detail the acquisition parameters of the NMR data in the appendix. I expect that high-field NMR would furthermore reveal the ¹⁵ND₄ spectrum, further simplifying analysis.

Authors' reply: We partially agree, however, signs of J-couplings are not revealed in high-field NMR while in ZULF NMR they can directly affect the spectra. Another big advantage is absence of susceptibility gradients.

Changes to the manuscript: We added a new Figure 5 as well as rewrote parts of the text dealing with comparison between ZULF and high-field NMR.

With respect to the pulse-length analysis, details of how the dependence is calculated should be added to the appendix. In Fig. 5, how are the black data points in panel A calculated from the spectra shown in Fig. 5B? For example the ¹⁵NH₄ peak at 110 Hz 10 us seems to be bigger than the (inverted) peak at 15 us, yet the data points in panel A indicate 1.5 and -1.5, respectively.

Authors' reply: We agree with these points. We now change the logic of presentation and extract the signs from running simulations of the spectra. A separate file containing simulation code in Mathematica is added as an SI file.

In the conclusion there is a statement that ¹⁴N in semi-symmetric environments can be used as tracer for /in vivo/ ZULF NMR studies without hyperpolarization. Even if the complete blood volume of an animal was replaced with a concentrated solution of ammonium chloride the filling fraction / signal would be only one tenth of what the authors get now. So how could this work?

Authors' reply: We have modified the text. However, a factor of 100 enhancement in sensitivity is possible, hyperpolarization can give another factor of 10,000. We are taking the small steps in the desired direction.

Minor points / suggestions::

p. 4: It is very clear from the appendix what the authors mean by "the hierarchy of spin-spin interactions", but it would probably help the reader to explain this concept in the main text.

Authors' reply: We agree, this is now clarified.

"We extract ^{15}N - ^1H , ^{14}N - ^1H spin-spin coupling values : replace comma with and

p. 5: add articles various spectral lines in /the/ J-spectrum, ... We additionally discuss /the/ possibility

p.6 We start from the analysis of the ZULF NMR spectrum of [^{15}N]-ammonium /cation/

p. 9 /in/ the same container

p. 12 ... upon increasing /the/ number of deuterium atoms

p. 12 ... Indeed, in a simple two-spin case, /the/ ZULF NMR signal...

p. 24 I have not checked the tables in detail, but I think the transition $3/2, 1$ to $1/2, 1$ should be moved down by one line in table 2. In the header of table A6 the Eigenstate symbols have to be changed to F and K_B

Authors' reply: We thank the Referee for the proof-reading, these typos are now fixed.

I am very much looking forward to see the authors' response and future work, as well as to learning where I was wrong! I wish them all the best for the future.

Benno Meier

REVIEWER COMMENTS

Reviewer #1 (Remarks to the Author):

I am satisfied with the modifications made by the authors.

Reviewer #2 (Remarks to the Author):

The revised manuscript represents a considerable improvement over the first version. Significant attention has been given to the determination of the ratio J_{15_NH} / J_{14_NH} .

However, questions regarding the determination of this ratio remain.

In Fig. 1a, the J_{14_NH} resonances (blue lines) are noticeably shifted to higher frequencies when compared to the experimental data (in black). How are these blue lines calculated? Would a direct fit of the experimental spectrum lead to a J_{14} / J_{15} ratio that is consistent with the gyromagnetic ratio?

The values of the direct fit that is mentioned in the SI should also be included in Tab S2 of the SI.

The caption of Fig. 2 should specify what the red and blue lines are in Fig. 2b. Reading the SI, one gets the impression that these should be Lorentzians, but they are noticeably skewed.

While the data partitioning provides good statistical error estimates, there is ample potential for systematic error. I am in particular concerned about the multiparametric baseline fitting process described in the SI. How robust are the results if the spectral windows for fitting the baseline are changed?

The experimental raw data should be deposited in a publicly accessible repository, preferably along with a notebook that implements the fitting described in the SI.

The outlook section is more moderate now, but still reads as if *in vivo* ZULF studies are now possible. This section requires a sentence stating clearly that the sensitivity of ZULF is too low to perform such experiments.

Other, minor points:

There is a statement in the manuscript that the positions of ZULF J-peaks are unaffected by digitization rate. Within the context the statement reads as if the position is also unaffected by uncertainty in the digitization rate. This is certainly not correct.

Figure 4 should use the same linewidth for the simulation and the experimental data.

In Fig. 5, why are there only 4 combinations of signs for the 3 J-couplings. Should there not be eight? Can the cases --- and --+ be discriminated based on the experimental data?

I recommend the manuscript for publication once the open questions regarding the determination of J_{14_NH} / J_{15_NH} have been addressed convincingly.

Benno Meier

Response to Referees

Authors' reply

Reviewer #1 (Remarks to the Author):

I am satisfied with the modifications made by the authors.

Authors' reply: We thank the referee.

Changes to the manuscript: None.

Reviewer #2 (Remarks to the Author):

The revised manuscript represents a considerable improvement over the first version. Significant attention has been given to the determination of the ratio J_{15_NH} / J_{14_NH} .

However, questions regarding the determination of this ratio remain.

In Fig. 1a, the J_{14_NH} resonances (blue lines) are noticeably shifted to higher frequencies when compared to the experimental data (in black). How are these blue lines calculated? Would a direct fit of the experimental spectrum lead to a J_{14} / J_{15} ratio that is consistent with the gyromagnetic ratio?

Authors' reply: we thank the Referee for spotting the shift. The solid line represented a fit with the old model used in the first version of the paper. Statistical analysis for extracting J_{15}/J_{14} ratio is now presented in Figure 2 and is completely updated. We should note, however, that the resulting value lies within the range consistent with Figure S5. The "direct fit" is also performed, and the positions of the peaks can be found in the 1st row of Table S2. The observed phenomenon still cannot be explained by taking a ratio of the gyromagnetic ratios alone.

Changes to the manuscript: Figure 1 was updated to remove the old fitting. Supporting Information was updated to include Figure S5 and Tables S3-S4.

The values of the direct fit that is mentioned in the SI should also be included in Tab S2 of the SI.

Authors' reply: We are afraid there is some confusion because, in fact, one can see the direct fit parameters marked with a star for clarification.

Changes to the manuscript: None.

The caption of Fig. 2 should specify what the red and blue lines are in Fig. 2b. Reading the SI, one gets the impression that these should be Lorentzian, but they are noticeably skewed.

Authors' reply: We thank the reviewer for the attention to detail. Red and blue colors are used to represent simulated spectra for ^{15}N and ^{14}N -isotopologues, respectively. The lines are not expected to be Lorentzian because what is shown is the absolute value (magnitude mode) of the complex Fourier data. We refer to this in the caption

"(b) Zero-field NMR spectrum of [^{14}N]- and [^{15}N]-ammonium (50:50 mixture, quadrature-detected magnitude mode, average of 36000 scans)",

We refer the reader interested in details of the fitting to the SI:

"It is important to note that the standard phase correction typically applied to high-field NMR spectra (0th and 1st order) proved insufficient for achieving universally positive absorptive lines in the ZULF NMR spectra. To address this, we implemented an interpolation of a frequency-dependent phase function $\varphi(\nu)$ enabling an absorptive phase across all relevant NMR peaks. The reason of this phase acquisition is currently under investigation. This modeling approach ensured the absence of baseline distortions introduced by high-order phase corrections."

Changes to the manuscript: The following line is added in the caption of Figure 2:

"ZULF NMR peaks of the [^{14}N]- and [^{15}N]-isotopologues are labeled with blue circles and red squares, respectively."

While the data partitioning provides good statistical error estimates, there is ample potential for systematic error. I am in particular concerned about the multiparametric baseline fitting process described in the SI. How robust are the results if the spectral windows for fitting the baseline are changed?

Authors' reply: The reviewer's insight prompted us to thoroughly revisit the analysis and test different options of removing the baseline from the ZULF NMR spectra. Results after carrying out different baseline removal procedures varied beyond the statistical error bar. Consequently, we used the standard deviation of these results to create a new "systematic error bar" representing the uncertainty introduced by the analysis procedure (**Figure 2c**).

As a result of the new error estimates, it is inconclusive whether the ratio of J -couplings differs when different peaks in the spectra are used, as the data values fall within the expanded error bar. References to such statements have been removed from the manuscript.

Changes to the manuscript:

- The supplementary information now contains the section "Systematic error of analysis procedure";
- Figure S3 has been added to the SI containing three illustrative examples showing different fitting results when using different baseline cleaning procedures;
- Table S3 and S4 were added to the SI summarizing all the different options used to test the robustness of the procedure;

- Figure 2 has been updated in the main text.

We additionally cited in the SI literature referencing to systematic errors induced by processing.

The experimental raw data should be deposited in a publicly accessible repository, preferably along with a notebook that implements the fitting described in the SI.

Authors' reply: We thank the referee and are happy to comply.

Changes to the manuscript: The files "ZULF_Baseline_Removal.py" and "ZULF_NMR_simulations.nb" have been added as supplementary material.

The outlook section is more moderate now, but still reads as if *in vivo* ZULF studies are now possible. This section requires a sentence stating clearly that the sensitivity of ZULF is too low to perform such experiments.

Authors' reply: We prefer not to speculate on whether or not *in vivo* ZULF NMR application will become a widespread clinical reality, however, we point out that an analysis on the available sensitivity limits of ZULF NMR detection coupled with hyperpolarization was recently performed.¹

Changes to the manuscript: None.

Other, minor points:

There is a statement in the manuscript that the positions of ZULF J-peaks are unaffected by digitization rate. Within the context the statement reads as if the position is also unaffected by uncertainty in the digitization rate. This is certainly not correct.

Authors' reply: We thank the reviewer for prompting us to be more precise in our wording. Indeed, an uncertainty in the acquisition rate would result in an uncertainty in total acquisition time and therefore bias the frequency bin, introducing a systematic error.

Changes to the manuscript: We have updated the text as:

"While similar analysis could in principle be performed using conventional high-field NMR, magnetic field drifts over long experimental time window and bounds on clock precision would necessitate additional post-processing which could introduce systematic errors while in ZULF NMR positions of the *J*-peaks are unaffected by field fluctuations and uncertainties associated with the demodulation of the reference clock in high-field NMR."

Figure 4 should use the same linewidth for the simulation and the experimental data.

Authors' reply: We agree with the comment of the reviewer. As expected, due to the narrow linewidths readily obtainable in ZULF NMR, in order to match the

experimental data, the simulations in Figure 4c and 4d have different linewidths: 0.5 Hz and 2 Hz, respectively. We would like to clarify that the linewidths in Figure 4a and 4b are indeed 0.5 Hz and 2 Hz. The reason why in (c) and (d) the simulation appear to have a larger amplitude is because no scaling has been made according to the deuterium fraction concentration of Figure 3.

Changes to the manuscript: None.

In Fig. 5, why are there only 4 combinations of signs for the 3 J-couplings. Should there not be eight?

Authors' reply: We thank the reviewer for the thoroughness. It is not possible to distinguish a global sign change in the spectra, therefore, e.g., simulations +++ and -- appear identical. As a consequence, half of the 8 possibilities are redundant and are not shown.

To illustrate this point below, one can see that of the 8 simulations in blue, the following are identical: 1st ↔ 8th, 2nd ↔ 7th, 3rd ↔ 6th, and 4th ↔ 5th.

Changes to the manuscript: For clarification we have added the following text in the caption:

“Given our insensitivity to global sign changes, we show only the 4 non-redundant permutations from the total 8 possible combinations of J -coupling signs.”

Can the cases --- and --+ be discriminated based on the experimental data?

Authors' reply: We thank the reviewer for this question. Indeed, those cases can be distinguished by inspecting the simulations of Figure 2a: “the wrong” case would not be able to show this good agreement between simulation and experimental data.

Changes to the manuscript: None

I recommend the manuscript for publication once the open questions regarding the determination of J14_NH / J15_NH have been addressed convincingly.

Benno Meier

- (1) Barskiy, D. A.; Blanchard, J. W.; Budker, D.; Stern, Q.; Eills, J.; Elliott, S. J.; Picazo-Frutos, R.; Garcon, A.; Jannin, S.; Koptug, I. V. Possible Applications of Dissolution Dynamic Nuclear Polarization in Conjunction with Zero- to Ultralow-Field Nuclear Magnetic Resonance. *Appl. Magn. Reson.* **2023**, *54* (11–12), 1221–1240. <https://doi.org/10.1007/s00723-023-01592-1>.

REVIEWERS' COMMENTS

Reviewer #2 (Remarks to the Author):

The authors have adressed the open questions and I recommend the paper for publication as is.

Benno Meier